# Maize Silage Pretreatment via Steam Refining and Subsequent Enzymatic Hydrolysis for the Production of Fermentable Carbohydrates

**DOI:** 10.3390/molecules25246022

**Published:** 2020-12-19

**Authors:** Malte Jörn Krafft, Olga Frey, Katrin U. Schwarz, Bodo Saake

**Affiliations:** Department of Chemical Wood Technology, University of Hamburg, Haidkrugsweg 1, 22885 Barsbüttel, Germany; malte.krafft@uni-hamburg.de (M.J.K.); olga.frey@uni-hamburg.de (O.F.); katrin.schwarz@uni-hamburg.de (K.U.S.)

**Keywords:** maize silage, ensiled maize, pretreatment, steam refining, enzymatic hydrolysis, carbon balance, biorefinery

## Abstract

Maize, also called corn, is one of the most available feedstocks worldwide for lignocellulosic biorefineries. However, a permanent biomass supply over the year is essential for industrial biorefinery application. In that context, ensiling is a well-known agricultural application to produce durable animal feed for the whole year. In this study, ensiled maize was used for steam refining experiments with subsequent enzymatic hydrolysis using the Cellic^®^ CTec2 to test the application possibilities of an ensiled material for the biorefinery purpose of fermentable carbohydrate production. Steam refining was conducted from mild (log R_0_ = 1.59) to severe conditions (log R_0_ = 4.12). The yields were determined, and the resulting fractions were characterized. Hereafter, enzymatic hydrolysis of the solid fiber fraction was conducted, and the carbohydrate recovery was calculated. A conversion to monomers of around 50% was found for the mildest pretreatment (log R_0_ = 1.59). After pretreatment at the highest severity of 4.12, it was possible to achieve a conversion of 100% of the theoretical available carbohydrates. From these results, it is clear that a sufficient pretreatment is necessary to achieve sufficient recovery rates. Thus, it can be concluded that ensiled maize pretreated by steam refining is a suitable and highly available feedstock for lignocellulosic biorefineries. Ultimately, it can be assumed that ensiling is a promising storage method to pave the way for a full-year biomass supply for lignocellulosic biorefinery concepts.

## 1. Introduction

Biorefinery concepts focusing on the sustainable production of energy or platform chemicals are highly discussed in academic circles of green and sustainable chemistry [1,2,3]. The main drivers are the enhancement of energy security, the support of rural landscapes by value-added technology and jobs, the slowdown of greenhouse gas emission, and the prevention of toxic compounds released in the environment [4]. The general framework condition and challenge for such concepts is the replacement of fossil-based resources [5]. Another main aspect is the dealing with social debates about food production, agricultural land consumption, and the usage of biomass for fuels or other chemistry processes, which is also known as “dinner plate or fuel tank” discussion [6]. Due to these requirements, the focus of research efforts in this field of work has been laid on new developments with lignocellulosic biomass (LCB) and the biomass supply [7,8,9].

Under the umbrella term LCB, it is possible to differentiate terms that are more practical. Therefore, LCB can be divided into wooden biomass and lignified biomass from monocotyledons, such as bamboo, bagasse, or straw and energy crops [10]. While the usage of wooden, virgin fibers for biorefinery competes with other wood-based products, the utilization of non-wood biomass, such as agricultural residues or natural non-wood plant fibers, is reported as beneficial due to the high availability, the open structure of the raw material, and its low price level [6].

One of the most worldwide available grains is maize, which is also called corn (*Zea mays* L.) [11]. Recent estimates of the Food and Agriculture Organization of the United Nations (FAO) show a corn production of 1,147,621,938 tons worldwide in 2018 [12]. In this context, some authors discuss prediction equations for the amount of corn straw, also called corn stover, based on the corn kernel to straw ratio of 1:0.88 [13], 1:1 [14,15], or 1:1.3 [16]. Using these proportions, a worldwide production of whole maize plant biomass, without roots, of around 2.15–2.64 billion tons can be estimated for 2018, which shows the high availability and relevance of this raw material.

Although many processes for the production of monomeric carbohydrates from LCB are based on enzymatic methods, LCB is known as naturally recalcitrant against microbial or enzymatic degradation due to its stable matrix, containing cellulose, hemicellulose, and lignin [17,18]. Thus, it is reported that a pretreatment step is necessary for processing LCB in biorefineries [19,20,21]. The main pretreatment objectives are to overcome the lignin barrier, to disrupt the crystalline structure of the cellulose chain, and thus, to further increase the porosity and to enhance the enzymatic accessibility of LCB [20,22]. All of the described aims are important for the following enzymatic hydrolysis (EH) and the sufficient production of monomeric carbohydrates by reducing the limiting factors of EH [23]. Nonetheless, EH is known as specific to cellulose and hemicelluloses, which are representing the main carbohydrates in LCB, with almost no formation of (inhibitory) by-products [24,25]. Therefore, the produced fermentable monomers are important building blocks for sustainable platform chemicals [3,26,27]. The most promising platform chemicals were firstly defined by Werpy et al. [28], also known as “DOE Top 10” [29], and they were updated and honed by several authors [2,29,30].

With respect to the pretreatment methods, it is possible to categorize the different processes into biological, chemical, physical, and thermal pretreatments [19]. In addition, physicochemical processes are described in the literature, such as steam explosion, ammonia fiber explosion, or CO_2_ explosion [22]. Steam explosion and a physical pretreatment of ensiled maize is described by Schober [31]. Further hydrothermal pretreatments [32,33] are described in the literature. Regarding plant engineering, all of these methods show as connecting elements nearly no necessary recovery, low or no chemical input, low process costs and common technology steps [22,34,35]. However, also methods without a separation of the solid and liquid fraction are described in the literature to reduce operational costs [36].

An additional or alternative physicochemical process is described as steam refining, which is characterized by a steaming step with a subsequent refining process [37]. This process is described by different authors for the production of chemical pulp [38], for the pretreatment of LCB, such as poplar wood [39,40], corn stover [41], spruce wood [42], and spruce forest residues [43]. Furthermore, possibilities for the utilization of alkaline-extracted steam-refining lignin in phenol-formaldehyde resins [44] and fractionation approaches for the recycling of waste medium-density fiberboards (MDF) by steam refining are reported [42]. The steam-refining process was also compared with the steam explosion processes by Schütt et al. [45].

Despite all the described pros and cons of LCB and its performance in different conversion processes, one of the primary challenges for LCB biorefineries is still the constant biomass supply during the year [46,47,48]. Due to these limitations, some biorefinery concepts utilizing more than one substrate [47] and concepts for a full year of delivery [49] are reported. Furthermore, the role of storage possibilities and the full-year delivery of the biomass is discussed in the literature [50]. A well-known method for full-year storage and delivery is ensiling of the substrate [51,52,53], especially for maize [54], corn stover [55], or LCB [56].

In a previous study, steam refining as pretreatment for corn stover was investigated [41]. The aim of the present study was to study an ensiled maize product due to the beneficial storage properties for further research. Due to the resulting high storage stability, ensiling of the raw material can be important for various industrial biorefinery applications with full-year biomass supply. Yields of the solid fiber and the liquid extract fraction after steam refining were determined. The achieved fibers were subjected to EH, and the extract fraction was characterized. Finally, process and mass balances were calculated based on the prior experiments and achieved results.

## 2. Results and Discussion

### 2.1. Raw Material Characterization

In order to give detailed information about the raw material, the anatomical composition was determined gravimetrically after sorting (Table 1). As the raw material was already shredded, husks were determined together with the leaves. Fines ≤ 4 mm were determined by sieving.

For harvested maize at the late dough-ripe stage, 61% cobs with kernels, 20.3% stalks, 17.3% leaves, and 1.5% husks are reported [57]. Daccord et al. [58] report about 58% cobs including kernels and husks, 25% stalks, and 17% leaves. Therefore, it can be shown that the amount of husks and leaves in the used material corresponds well to the aforementioned publications.

A regular chopping size of 5 mm is reported [59]. Therefore, the limit for fines was set at less than or equal to 4 mm. As shown in Table 1, a not classifiable, fine fraction occurs after manual sieving. Bruni et al. [60] show also a significant fine fraction for maize silage. Reasons for the fine parts can be a smaller chopping size of the chopper or other technical aspects during harvest. With view to the data, it is clearly visible that the values for stalks and cobs including kernels are lower than stated in the literature. Therefore, it can be assumed that the fine fractions contain, beside small leaf fragments, mainly fragments of the brittle parts of the plant, such as kernels, stalks, and cobs, and these values are underestimated due to the occurring fine fraction.

For further understanding and characterization of the raw material, the extractives and carbohydrates were determined (see Table 2).

For the detection of carbohydrate losses in the ASE, the carbohydrate analysis was performed with and without extraction. The carbohydrate content of the non-extracted material was determined by two-step acidic hydrolysis. In that context, Thomsen et al. [32] report contents of about 51.7% glucose, 19.5% hemicelluloses, and 16.6% lignin for non-extracted maize silage after acidic hydrolysis. This is in good accordance with the measured data. Cellulose, xylan, and Klason lignin are reported by Xu et al. [61] in nearly similar values. The determined ash content is also in good accordance with the present Weende analysis. It must be remarked that without extraction, glucose from starch and cellulose is detected together due to the analytical method. However, some degradation of the starch can be expected due to the strong acidic hydrolysis. In addition, the present animal feed analysis for the used raw material reports about an acidic detergent lignin (ADL) value of 1.6%. With respect to the reported values, the measured acidic insoluble residues, which are reported to be mainly analogous to Klason lignin [62,63], seem to be overestimated due to present proteins in the maize silage. Notwithstanding, an extraction step is regularly used before raw material analysis and is reported as important due to the possible precipitation of extracts or a limited access to the substrate resulting in incorrect lignin contents or too low carbohydrate yields [64]. Therefore, the raw material was extracted, and the raw material composition was additionally tested for the extracted material.

After extraction, a loss of glucose is visible in comparison with the non-extracted material. It can be assumed that the high values for the water extracts are a possible reason for that observation and that hot water-soluble polymers, such as starch, are detected partly in the water extract. This suggestion can be supported by the observation of starch paste in the water extract.

Due to the reported high starch contents and the possible degradation during the two-step acidic hydrolysis, an enzymatic degradation of the starch prior to an acidic hydrolysis was tested (Table 2). Afterwards, a starch content of 38.6% could be determined by enzymatic starch degradation. This result is in good accordance with literature values for starch and total carbohydrates given by Schober [31] and with results of the Weende analysis with a reported starch content of 37.8%. Furthermore, also the values for cellulose and the hemicelluloses show more similarities [31]. It can be concluded that the hot water extraction solved a part, but not all, of the starch. Therefore, an enzymatic treatment with subsequent acidic hydrolysis is the best method in order to characterize the raw material and differentiate carbohydrates from starch and other polysaccharides. The conventional determination applied for LCB with an extraction prior to the acidic hydrolysis is not suitable for this raw material.

### 2.2. Fiber and Extract Yields after Steam Refining

After steam refining, yields of the solid fiber fraction and the liquid extract fraction were determined. Steam refining was conducted with severities of 1.59 up to 4.12 (Table 5). The determined fiber and extract yields are illustrated in Figure 1.

As illustrated in Figure 1, the fiber fraction shows a strong decreasing tendency with increasing severity. Starting with a fiber yield of around 50%, the yield decreases up to a minimum value around 30%. In contrast, the extract fraction increases continuously from 50% up to 72% at a severity of 3.65. A reduction to 63% occurs at the highest applied severity of 4.12. The total yield loss of around 7% can briefly be explained with degradation reactions (see Section 2.4).

Several authors discuss the yield development of fibers and extracts for different raw materials. Krafft et al. [41] also show a decrease of steam refined corn stover fiber yields above log R_0_ = 4. As obtained in the present data, the maximum extract yield is changing at severities higher than log R_0_ = 3.5 to a scattering or slight decreasing tendency. This might be an indication for increased degradation reactions. A decrease of the extract yields is also reported by Krafft et al. [41] for corn stover but at higher severities.

Xu et al. [61] report also fiber yields of around 46% for hydrothermal pretreated ensiled maize at a severity of 2.98 and decreasing yields with increasing severity. For steam-refined corn stover, fiber yields above 80% and extract yields below 20% at a severity of 2.77 are reported [41]. The high extract yields and the relatively low fiber yields, especially at low severities, show a completely different behavior of ensiled maize during steam refining in comparison with other substrates. A possible explanation may be the date of harvest or the degradation of the biomass during the acidic-based ensiling process with pH values below 4. Furthermore, hemicelluloses and starch may become water-soluble under steaming conditions. Even after steaming and refining at comparatively low conditions, around 50% of the maize components is either soluble or degraded to fines, which pass the sieve, and is therefore detected in the extract. Due to the preceding ensiling process, an attenuation of the pretreatment conditions seems possible. It is known from the literature that the step of pretreatment is the most expensive one within the whole process, including also enzyme production and EH [19,20,65]. In that context, the optimization and lowering of the heat and energy consumption for biorefineries are addressed in several publications [66,67].

### 2.3. Chemical Characterization of Fibers and Extracts

To gain detailed information about carbohydrates in the fiber and extract fraction and occurring degradation reactions, the carbohydrate composition of both fractions was determined (Table 3). Furthermore, 5-HMF, furfural (Figure 2), and organic acids were analyzed in the extract fraction to evaluate the concentration of inhibitors and degradation products.

Different tendencies can be shown for the raw material-based composition of the fiber fraction. With increasing severity, the glucose yield is slightly decreasing. The xylose content is stable up to a severity of 2.77. Afterwards, a decrease of xylose is visible for the fiber fraction. Arabinose and other carbohydrates are constantly decreasing, whereas the hydrolysis residue, analogs to Klason lignin, is lower compared to corn stover and nearly stable.

For a hydrothermal pretreatment of ensiled maize between severities of 2.98 and 3.68, Xu et al. [61] report fiber-based values for the solid fraction. By recalculating the values with the fiber yield to a raw material-based view, they report similar cellulose contents between 15.6% (log R_0_ = 2.98) and 14.4% (log R_0_ = 3.68). Furthermore, xylan contents between 9.5% (log R_0_ = 2.98) and 5.1% (log R_0_ = 3.68) are reported by recalculation. It is most likely that the solubility of the available starch at higher severities contribute to the high value of detected glucose, which is reported by Xu et al. for the extract fraction [61]. They report more than 70% of the theoretical available starch in the extract fraction. Only around 10% of the theoretical available starch remains in the fiber fraction [61]. The decreasing tendency for the hemicelluloses is known and is caused by degradation reactions and/or the preferential solubilization of hemicelluloses [41]. In that context, Krafft et al. [41] show glucose yields around 30% for the solid fraction of corn stover. Higher values are also reported for the xylose yield but also with the described decreasing trend with increasing severity. The stated composition of the fiber fraction is important for the knowledge about the theoretically available carbohydrates. These values will be compared in Section 2.5 with the carbohydrate contents after EH to gain information about the conversion ratio to monomers.

For the glucose content of the extract fraction, an increase was determined. The xylose content increases with severities from log R_0_ = 2.77 to 3.65. Hereafter, a turnaround to a decrease is shown. Other hemicelluloses are detectable in traces, following the xylose trend. Residues are scattering at comparable low values. Therefore, the development of the xylose content is in good accordance with the previous findings and the literature. At a severity higher than 2.77, xylose is dissolved and will be found in the extract fraction. At higher severities, the dissolved hemicelluloses will be degraded. Similar findings are reported in the literature [41,68]. However, due to the deviation of steam-refining extracts from other substrates, the comparable high glucose contents are of great interest. For steam-refining extracts of corn stover, lower glucose values are reported [41]. With that view, the high glucose contents in the extract fraction are remarkable. Reasons might be solubilized starch during the steaming reactions or degraded, fine biomass in the extract, which is passing through the used sieve, which had a mesh size of 50 µm. The determined carbohydrate composition of the extract fraction is important for further determinations of mass balances and the calculation of the optimal total carbohydrate recovery point. Therefore, these values will be used for further considerations in Section 2.5.

### 2.4. Degradation Products, Organic Acids, and pH Value

As reported in the aforementioned findings, degradation reactions of the carbohydrates during steaming at high severities are well known. Therefore, degradation products, which are partly known as inhibitors in later fermentation process steps [69,70,71,72], were determined. In that context, 5-HMF, furfural, lactic acid, acetic acid, formic acid, and the pH value were determined. The values for 5-HMF and furfural are illustrated in Figure 2.

As illustrated, the contents of 5-HMF and furfural in the extract increase with increasing severity. The increase of the furan content is described in the literature as a result of the degradation of hexoses and pentoses under severe conditions [40,73,74,75,76,77]. For 5-HMF and furfural, an increase is visible at a severity around 2.94. At higher severities, close to a severity of 4.0, the increase of the 5-HMF content is much stronger than for the furfural content. For ensiled maize experiments, comparable results of 0.17–0.27% furfural and 0.04–0.1% 5-HMF content are reported at a severity of 3.5, although the heating time was much longer [33]. Nevertheless, a negative effect due to inhibitory effects is reported for the methane production of hydrothermal pretreated maize silage, although the inhibitor values are low [33].

In comparison with corn stover, steam-refined maize silage show higher furan contents [41]. Nonetheless, the loss of carbohydrates illustrated in Table 3 can be partly explained by degradation reactions and the increasing furan contents.

It is also known that 5-HMF and furfural can be degraded under severe conditions to formic acid. 5-HMF can further be degraded to levulinic acid. In addition, the liberation of acetic acid caused by hemicellulose degradation is described and is also reported in the term of autohydrolysis [78]. Furthermore, the natural occurrence of lactic acid in ensiled products is known. Due to these effects, a lowering of the pH is often observed [70].

Therefore, the lactic acid content is nearly stable from 5.87% at the lowest (log R_0_ = 1.59) up to 4.79% at the highest severity (log R_0_ = 4.12). Acetic acid is also nearly stable between 1.54% and 1.82%. The pH value show only a slight reduction from 4.2 to 3.9. However, formic acid shows a slight increase from 0.06% to 0.28%.

As described, the values for lactic acid are nearly stable. It is known from the Weende analysis prior to steaming that the raw material contains 4.9% lactic acid from the ensiling process. Therefore, it can be assumed that the lactic acid in the extract fraction originates from the raw material. During steaming, the lactic acid was dissolved completely without any forming or degradation reactions. With view to the acetic acid, slightly higher values are reported for corn stover steam-refining extracts [41]. For the formation of formic acid, an increasing trend is visible in the present data. A degradation of the furans at severe conditions is a possible explanation for these phenomena. The measured pH values are in contrast to reported pH values of corn stover extracts [41] after steam refining. However, the knowledge of acetic acid liberation and the occurring acetic acid contents is important for conclusions about autohydrolysis. Schütt [79] reports on high contents of acetyl groups for hardwood xylan and therefore, an occurring autohydrolysis during steaming. In the case of softwood, only limited autohydrolysis takes place due to low acetyl group contents. Therefore, the usage of acidic catalysts, such as SO_2_, is indicated for softwood. For poplar wood, acetic acid values of 1% to 3% are presented [45], whereas values below 1% are described for analogous experiments with softwood without catalysts [42]. Therefore, with values higher than 1%, an occurring autohydrolysis can be suggested for ensiled maize, and the usage of acidic catalysts is not needed, which is also beneficial in terms of cost considerations.

### 2.5. EH of Pretreated Fiber Fraction

After steam refining pretreatment, EH was conducted on the fiber fraction to find further details for the influence of the severity on the enzymatic saccharification. The results, expressed as the recovery of the theoretical available glucose and xylose after EH, are illustrated in Figure 3. Furthermore, enzymatic hydrolysis without steam refining was conducted as blind value for the raw material. In this case, a carbohydrate recovery of 55.1% was achieved, which represents 30.1% glucose and 1.6% xylose in absolute values based on raw material.

As illustrated in Figure 3, the EH yield of the fiber fraction is increasing steadily with increasing severity. Values of around 50% carbohydrate recovery were obtained at the lowest severity of 1.59, and a recovery of 100.9% was determined after the pretreatment under the most severe conditions (log R_0_ = 4.12). Results higher than 100% are reported by other authors, too [41,61]. One possible explanation is the determination of the theoretical carbohydrates by acidic hydrolysis, which results into some carbohydrate degradation. Compared to that, EH is more selective without carbohydrate degradation reactions. Therefore, an efficient enzymatic hydrolysis can result in higher values than after acidic hydrolysis.

Comparable enzymatic hydrolysis experiments using Cellic^®^ CTec2 with steam-treated corn stover showed similar developments. At the found optimal severity for corn stover (log R_0_ = 3.95), an EH carbohydrate recovery of 86.4% was described [41]. However, also at the highest severity, they did not reach 100% recovery. The presented data show that it is possible with ensiled maize to reach 100% of the theoretical carbohydrates in the fiber fraction as monomers after pretreatment, but a pretreatment step is necessary to reach that goal.

To find the optimal pretreatment condition, Figure 4 show the comparison of the carbohydrate yields after EH of the fibers and from the extract fraction. It can be seen that the highest amount of carbohydrates can be achieved after pretreatment at a severity of 3.65. After that point, the decreasing carbohydrate yield of the extract fraction influence the total yield. This is in good accordance with e.g., values for corn stover, where the optimum is reported at a severity of 3.95 [41].

The present finding for ensiled material shows that a lowering of heat and temperature is possible, which can be a benefit for further economic considerations aiming at the reduction of operational costs. Due to the possibility of ensiling in bunker silos, no further buildings for storage are necessary, which also reduces the operational costs in case of costs for building constructions. However, ensiling enables also a safe storage for a long time and allows a full year of biomass supply, which is reported by several authors [80,81].

With view to the whole process time consumption, it can be stated that ensiling does not extend the process. Rather, it is a beneficial storage method for the biomass supply. While ensiling, non-ensiled or partly ensiled biomass is available for the process. It is known that steam refining or explosion is also working with fresh material, such as non-ensiled corn stover [41] or other biomass [82]. However, the present data show an appropriate way for storage with further beneficial influence on the enzymatic hydrolysis yields.

### 2.6. Mass and Carbon Balance

To get an overview of the process, a mass balance was made for the found optimal severity. In Table 4, the carbohydrates, hydrolysis residues, organic acids, and furans are listed for the extract and fiber fraction. For comparisons, the raw material analysis is provided as well. The carbohydrates determined in the extract and fiber fraction are up to 95% of the carbohydrates in the raw material, which is a very high recovery and conversion rate. All compounds identified in the extract and fiber fraction amount to 87.1% based on raw material. The analytical material analysis results into 89.7% of identified compounds. The deficit in the raw material analysis is due to several reasons, such as inorganic material, analytical losses, and unidentified components, such as extractives, which are unidentified in the characterization method. To further evaluate the efficiency of the process, a carbon balancing was performed, since the carbon efficiency is a fundamental principle of green chemistry [83,84,85,86,87]. Therefore, approaches for carbon balances in biorefineries are described in the literature [88].

As illustrated in Table 4, the highest amounts of carbon were determined in the extract fraction. Combining the extract and fiber fraction, 46.6 g of carbon in 100 g of material was found compared to 46.8 gC/100 g in the raw material. This represents a recovery rate of 99.6%. Around 0.2 gC (0.4%) of the theoretical available carbon can be most likely attributed to losses in the vapor phase. The high carbon yields in the extract can be explained by the high reported carbohydrate yields in this fraction. Furthermore, the components, which might not be detected in the previous analysis, will be mostly contained in the extract. With respect to the literature, Beltrame et al. [89] show a conversion of up to 30% to volatile compounds for steam explosion of wheat straw at a severity of 4.24. Kaar et al. [75] report about a volatile loss of up to 22% for the steam explosion of sugarcane bagasse. In contrast, Turn et al. [90] report for a dataset of 77 steam explosion conditions between severities of 3.7 and 4.3 only small amounts of around 2% of gas phase carbon at the highest and around 0.5% at the lowest severity of 3.7. Candidates for compounds in the volatile fraction are reported as furfural, formic acid, or acetic acid for steam refining of corn stover, whereas no 5-HMF was found [91]. Furthermore, methane, ethane, propane and, as the main carbon-containing compound in the vapor phase, carbon dioxide, are reported in the literature for the gas phase after steam explosion of banagrass, which is a variety of elephant grass (*Pennisetum purpureum*
Schumach.) [90]. Therefore, the found small amounts of carbon loss are in accordance with the results presented by Turn et al. [90]. It is a positive outcome that the found optimal condition has a low severity, which is limiting the losses of volatile organic compounds.

## 3. Materials and Methods

### 3.1. Raw Material

The raw material was harvested as full plant (without roots) with a commercial maize chopper in 2018 near Brietlingen, Lower Saxony, Germany (4 m above NHN). It was harvested at the dough-ripe stage, which represents the optimal time to harvest with view to the nutritive value and the ensiling properties [92]. After harvesting, the biomass was compacted and covered airtight for ensiling. After three months, ensiling of the biomass was completed. The sampling was conducted in airtight barrels directly on the dairy farm. The dry matter content of the material was 37.9%. The barrels were stored frozen (−18 °C) to stop further biomass conversion. For steam refining, the samples were defrosted prior to the experiment, and the moisture content was determined for each batch in triplicate at 105 °C ± 3 °C according to TAPPI T 264 cm-07.

The ash content of the raw material was determined for comprehensive raw material analysis according to TAPPI T 211 om-16/ISO 1762:2019 at 525 °C ± 25 °C. The determination of the extractive compounds was conducted by three Accelerated Solvent Extraction (ASE) steps at ≤1 mm milled material. The first two extractions were performed for 10 min at 10 MPa with an ASE 350 (Thermo Scientific^®^ Dionex^®^, Waltham, MA, USA) with petrol ether and acetone/water (9:1), both at 70 °C. Due to the occurring starch paste at higher temperatures, the hot water extraction was performed in two steps. About 3 g of the pre-extracted residue was extracted in a first step for 45 min in 700 mL demineralized water. Afterwards, an extraction thimble was filled with the sample, and the extraction was continued with a Soxhlet apparatus for 3 h with 600 mL demineralized water. For all extraction steps, 10 mL of the extract was removed, dried, and the extract content was determined gravimetrically. The extractive-free raw material was air-dried for further carbohydrate determination by acid hydrolysis.

For the determination of the starch content, 300 mg of theoretical dry raw material was treated with an enzymatic solution containing 0.05% amyloglucosidase and 0.05% amylase in a pH 4.7 ammonium acetate buffer. The raw material was treated for 24 h at 46 °C. Afterwards, the reaction was stopped, and the samples were filtered through a G4 sintered glass frit. The filtrate was further used for glucose detection via borate–anion exchange chromatography (AEC).

For carbohydrate detection, a two-step sulfuric acid hydrolysis according to Lorenz et al. [93] was performed. After the second hydrolysis, the flasks were cooled down to room temperature, filled up with demineralized water, and filtered to a G4 sintered glass frit. For further analysis of the carbohydrates (see Section 3.4), the undiluted filtrates were collected. The residues in the frit were washed, dried overnight at 105 ± 3 °C, and weighted gravimetrically to determine the amount of acid-insoluble residue, which is comparable with Klason lignin [62]. For raw material characterization, the acid soluble lignin was measured additionally at 205 nm using 2.3% sulfuric acid. Additionally, a commercial animal feed analysis, also called Weende analysis, was done by Eurofins Agro, Wageningen, Nederlands, for comparative aspects.

### 3.2. Pretreatment Process

Steam refining was conducted with moist raw material with a dry matter content of around 35%. A value equivalent to 200 g of dry material was inserted into the steaming reactor (22 × 25 cm, Martin Busch & Sohn GmbH, Schermbeck, Germany) to make the process comparable. The severity factor (R_0_) was calculated according to Overend and Chornet [94] as the more common log R_0_ using Equation (1):(1)$$\log R_{0}=\log\;\left(t\;\times {\;e}^{\frac{\left(T-100\right)}{14.75}}\right).$$

The severity factor combines the time of pretreatment (t) in minutes with the pretreatment temperature in °C (T). With this factor, it is possible to generate a comparability for pretreatments with different durations and temperatures in the same dataset.

The steaming reactor was equipped with a four-blade refiner system, which was started in the last 30 s of pretreatment with 1455 rpm. Afterwards, the refining was stopped, and the pressure was released. According to Table 5, twenty experiments, including one blind value, were carried out at temperatures between 120 and 200 °C with different pretreatment durations. Furthermore, the data point at 160 °C and 20 min (log R_0_ = 3.07) was included to get an overview over the different influences of severity, temperature, and time in comparison with the data point at 170 °C and 10 min (log R_0_ = 3.06).

The liquid extract fraction and the solid fiber fraction were separated after steaming using a 50 µm sieve. The fiber fraction was dried for 10 min at 2800 rpm in a spin-dryer (Thomas Centri 776 SEK, Robert Thomas Metall- und Elektrowerke GmbH & Co. KG, Neunkirchen, Germany), homogenized for 10 min in a 20 L rotary stirrer (Hobart A20, Hobart GmbH, Offenburg, Germany), and finally, the weight was determined gravimetrically.

The extract fraction was also collected, weighted, and the pH value was determined. The dry matter contents of both fractions were determined for yield calculations. The dry matter content of the extract was determined by freeze-drying (Alpha 2–4 LSC; Martin Christ Gefriertrocknungsanlagen GmbH, Osterode am Harz, Germany), the dry matter content of the fibers was determined according to TAPPI T 264 cm-07 at 105 ± 3 °C. All samples were stored frozen afterwards to prevent degradation of the samples. A scheme of this process is illustrated in previous work [41].

### 3.3. Acid Extract and Fiber Fraction Hydrolysis

For carbohydrate determination of the fiber fraction, the above-mentioned two-step acidic hydrolysis (see Section 3.1) was conducted. The dried extract lyophilisates were hydrolyzed with a one-step dilute sulfuric acidic hydrolysis procedure, which was modified according to Lorenz et al. [93]. First, 100 mg of the lyophilisate was mixed with 10 mL of demineralized water in an ultrasonic water bath. After dilution of the extract, 1.8 mL of 2N H_2_SO_4_ was added. The suspension was hydrolyzed for 40 min at 120 °C and 0.12 MPa. Hereafter, the hydrolyzed samples were filled up and filtered through a G4 sintered glass frit. The undiluted filtrate was collected for further analysis. The residue in the frit was collected and determined as reported before.

### 3.4. Chromatographic Methods

Borate–anion exchange chromatography (borate–AEC) was used for monomeric carbohydrate detection after acidic hydrolysis and enzymatic hydrolysis. A Dionex^TM^ UltiMate^TM^ 3000 (Thermo Fisher Scientific^TM^, Waltham, MA, USA) was used with an anion exchange resin (MCI GEL^®^ CA08F, Mitsubishi Chemical, Tokyo, Japan). Two buffers with pH 8.6 and 9.5 based on potassium tetraborate/boric acid were used in different concentrations over the time after a post-column derivatization working at 65 °C. Carbohydrate determination was conducted via UV/VIS-spectroscopy at 560 nm. A more detailed explanation of this method is reported by Lorenz et al. [58].

Organic acids, such as lactic-, acetic-, and formic acid were detected by ion chromatography. The extracts were filtered, and 5 µL were separated with an IonPac^TM^ AS11-HC (2 × 250 mm) anion exchange column (Dionex, Sunnyvale, CA, USA), using a Dionex^TM^ ICS 2000 with a Dionex ASRS 300–2 mm suppressor. A flow rate of 0.38 mL/min^−1^ was adjusted, and a temperature of 35 °C was set. As eluent, potassium hydroxide (KOH) was used with concentrations between 1 and 70 mM over the time.

Furfural and 5-hydroxymetylfurfural (5-HMF) were detected by RP-HPLC. Therefore, 20 µL of fresh extract was separated at 25 °C for 80 min using an AQUASIL^TM^ C_18_ column (250 × 4.6 mm × 5 µm; Thermo Fischer Scientific^TM^, Waltham, MA, USA). A flow rate of 1 mL/min^−1^ was set, and two eluents (A: weak acidic water, 1 mM H_3_PO_4_; B: acetonitrile, C_2_H_3_N) were used in different concentrations over the time, starting with 97.5% A and 2.5% B and ending with 100% B. For detailed information about the concentration steps over the time, see Krafft et al. [41]. The wavelength for detection was set at 280 nm.

For carbon detection in the raw material and fiber fraction, a vario EL cube (Elementar Analysensysteme GmbH, Langenselbold, Germany) was used. Carbon in the extract fraction was detected via total organic carbon (TOC) according to DIN EN 1484:1997-08 by Eurofins Umwelt West GmbH. Carbon in the vapor phase was calculated as ΔgC between the raw material and fiber/extract values.

### 3.5. Enzymatic Hydrolysis (EH)

The EH of the pretreated fibers was conducted at a dry matter content of 4% with 300 µL cellulases (Cellic^®^ CTec2, 91 FPU/mL, Novozymes A/S, Bagsværd, Denmark) and 50 µL glucosidases (Novozyme 188, Novozymes A/S, Bagsværd, Denmark), with an β-glucosidase activity of 227 U/mL. For that purpose, one gram of dry material content of the pretreated substrate was hydrolyzed for 72 h at 45 °C. A pH value of 5.0 was adjusted using a phosphate citrate buffer. After hydrolysis, the samples were filled up to 250 mL in a volumetric flask with demineralized water and filtered through a G4 sintered glass frit. The undiluted filtrate was collected for carbohydrate determination using borate–AEC.

## 4. Conclusions

The present study and the results are open to several conclusions. It was found that the steam refining of ensiled maize leads to high extract yields due to a good enzymatic accessibility. Furthermore, high amounts of starch and low amounts of lignin were determined in comparison with corn stover.

However, it was found that a pretreatment step is necessary for a sufficient enzymatic hydrolysis of the fiber fraction. In comparison with untreated maize silage, it was possible to achieve nearly a doubling of the EH yields.

The optimum steam refining condition for the production of monomeric carbohydrates was set at a severity of 3.65. At this point, the optimum between carbohydrates from EH and the carbohydrates determined in the extract fraction was found with a total recovery of 62.6% of carbohydrates, which is based on raw material input, with low carbon losses in the gaseous phase.

Furthermore, ensiled maize was confirmed as a potential LCB for the production of monomeric carbohydrates after pretreatment. In addition to the benefit of a high availability and a full year of biomass supply due to the ensiling, the ensiling process seems to be beneficial for the further processes in comparison with other steam refined, non-ensiled substrates. Therefore, a recovery of around 100% of the theoretical available carbohydrates was reached for the fiber fraction after pretreatment.

Due to the knowledge that ensiling is not affecting the chemical composition of the substrates [95] and due to the results from the present study, ensiling seems to be a promising storage method for LCB biorefineries to achieve a full year of biomass supply by storage.

It seems to be promising for future work to test more LCB substrates, such as agricultural residues or rural wastes, on their ensiling potential and the carbohydrate recoveries after pretreatment and EH. A promising future candidate for further experiments might be ensiled corn stover, which represents an undervalued agricultural residue with synchronous good ensiling properties [96].

## Figures and Tables

**Figure 1 molecules-25-06022-f001:**
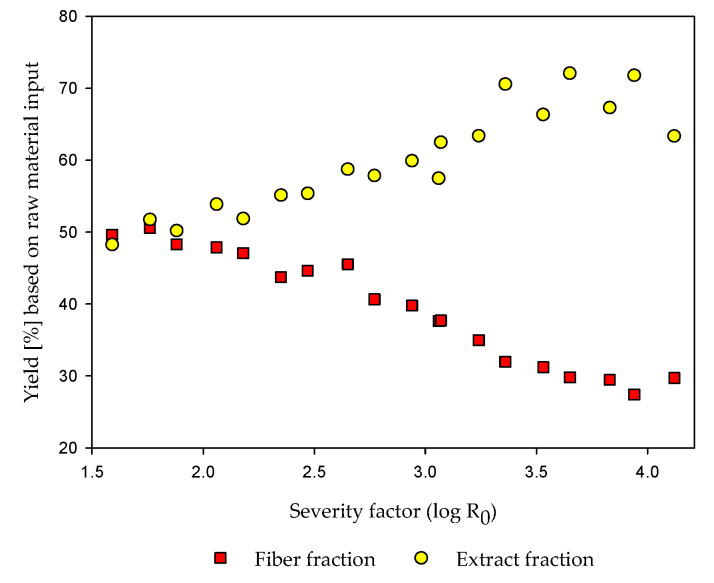
Fiber and extract yields after steam refining of ensiled maize.

**Figure 2 molecules-25-06022-f002:**
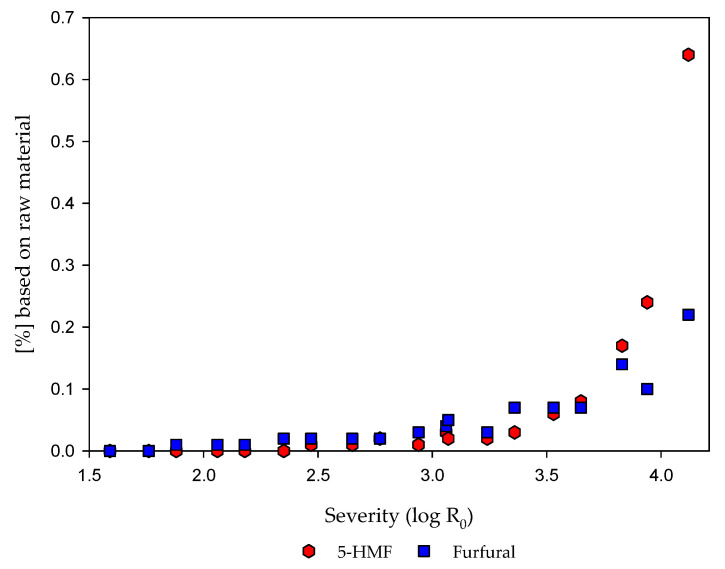
Measured 5-hydroxymetylfurfural (5-HMF) and furfural contents in the extract fraction based on raw material.

**Figure 3 molecules-25-06022-f003:**
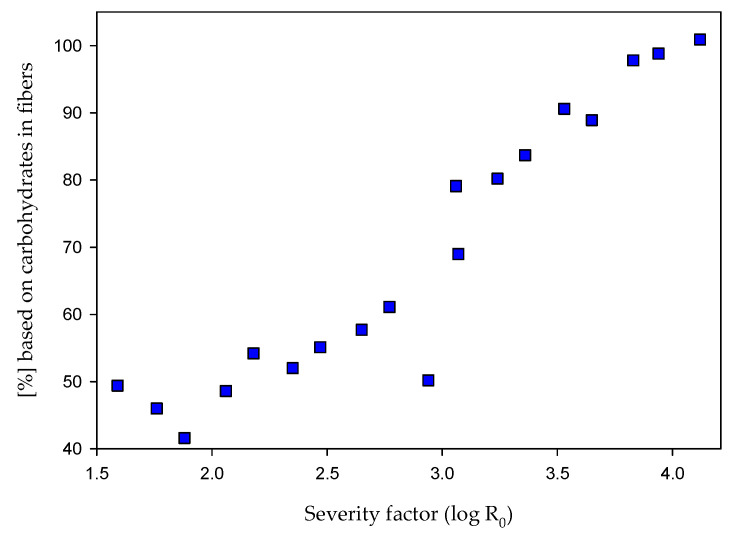
Results after enzymatic hydrolysis (EH) of the pretreated fiber fraction expressed as carbohydrate recovery compared to the theoretical available carbohydrates before EH.

**Figure 4 molecules-25-06022-f004:**
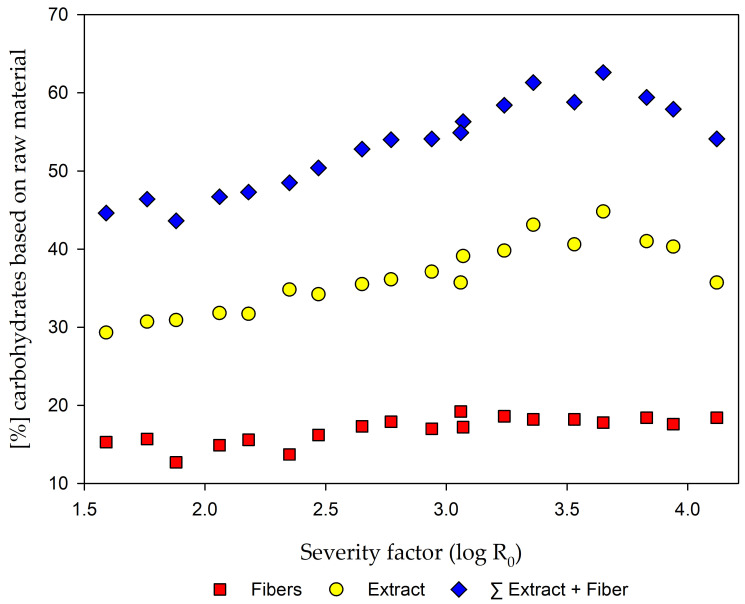
Comparison of carbohydrates after EH of the fibers, carbohydrates from the extracts, and the sum of both, all based on raw material.

**Table 1 molecules-25-06022-t001:** Fractions of the used raw material.

	Raw Material % (*w*/*w*)
Stalks	15.8
Kernels	17.7
Cobs	14.0
Leaves and Husks	23.8
Fines (≤ 4 mm)	28.7

**Table 2 molecules-25-06022-t002:** Chemical characterization of the raw material before and after extraction, and after enzymatic starch degradation prior to the two-step acidic hydrolysis. All values are calculated as percentage (*w*/*w*) based on non-extracted raw material.

		Non-Extracted	Extracted	Starch Determination
		% (*w*/*w*)		
Extraction	Petrol ether	n.d.	2.5	n.d.
	Acetone/Water (9:1)		6.6	
	Hot water		15.6	
	∑		24.7	
Starch		n.d.	n.d.	38.6
Carbohydrates	Glucose	51.9	43.8	18.3
	Xylose	12.3	13.7	11.0
	Arabinose	2.2	2.5	2.0
	Galactose	0.8	0.8	0.6
	Mannose	0.2	0.3	0.1
	Rhamnose	0.1	0.1	0.1
	∑	66.0	61.2	70.7
Lignin	acid-insoluble	11.1	9.3	11.7
	acid-soluble	3.2	2.6	1.5
	∑	14.3	11.9	13.2
Ash		3.49	3.52	n.d.
Organic acids *	Lactic acid	4.9	n.d.	n.d.
	Acetic acid	0.9		
Elements	C	n.d.	46.8	n.d.
	H		6.1	
	S **		0.6	

* Determined by Eurofins Agro. ** Value for sulfur is close to the detection limit and therefore only approximately correct.

**Table 3 molecules-25-06022-t003:** Carbohydrate and residue content for the fiber and extract fraction based on raw material input.

	Fiber Fraction					Extract Fraction				
log R_0_	Gluc	Xyl	Arab	Others *	Residue **	Gluc	Xyl	Arab	Others *	Residue
	[% *w*/*w*]					[% *w*/*w*]				
1.59	21.4	9.7	1.5	0.7	12.4	28.6	0.7	0.3	0.2	1.5
1.76	22.7	11.3	1.8	0.7	12.3	30.0	0.7	0.3	0.2	2.3
1.88	20.2	10.5	1.7	0.7	12.2	30.2	0.7	0.3	0.2	1.6
2.06	20.1	10.6	1.6	0.6	12.4	31.0	0.8	0.5	0.3	3.3
2.18	19.4	9.5	1.4	0.6	12.4	30.9	0.8	0.5	0.3	1.5
2.35	17.6	8.8	1.2	0.6	12.1	33.9	0.9	0.6	0.2	1.7
2.47	19.3	10.1	1.3	0.6	11.4	32.9	1.3	0.9	0.3	1.3
2.65	19.6	10.5	1.2	0.5	11.4	34.2	1.3	0.8	0.4	2.5
2.77	19.5	9.8	1.0	0.5	12.0	34.2	1.9	1.1	0.3	1.3
2.94	17.1	7.7	0.7	0.4	11.2	34.7	2.4	1.0	0.4	2.5
3.07	17.4	7.5	0.6	0.3	10.9	35.7	3.4	1.2	0.6	0.6
3.06	17.3	7.0	0.5	0.3	11.0	32.0	3.7	1.2	0.4	1.8
3.24	17.0	6.2	0.4	0.3	9.8	35.1	4.7	1.2	0.8	3.2
3.36	16.9	4.8	0.2	0.2	9.3	36.4	6.7	1.5	0.8	3.9
3.53	16.4	3.7	0.2	0.2	9.8	34.0	6.6	1.1	0.9	2.8
3.65	16.7	3.3	0.2	0.1	9.4	37.8	7.0	1.3	0.8	4.2
3.83	16.4	2.4	0.1	0.1	9.9	34.1	6.9	1.0	1.0	2.9
3.94	16.0	1.8	0.1	0.1	9.3	33.9	6.4	1.0	0.8	5.2
4.12	16.7	1.6	0.1	0.1	11.5	30.6	5.1	0.7	0.8	0.9

*: containing rhamnose, galactose, and mannose. **: containing the acid-soluble lignin and the acid-insoluble residue.

**Table 4 molecules-25-06022-t004:** Mass balance of carbohydrates (CH), residue, organic acids, furans, and the carbon balancing as gram carbon (gC) after steaming at the optimal point of 3.65.

	∑ CH	Residue	∑ Organic Acids	∑ Furans	∑ Carbon
	[%] Based on Raw Material				gC/100 g
Extract	46.9	4.2	6.17	0.15	32.0
Fiber	20.3	9.4	-	-	14.6
∑	67.2	13.6	6.17	0.15	46.6
Raw Material	70.7	13.2	5.8	-	46.8

**Table 5 molecules-25-06022-t005:** Experimental plan, including temperature, reaction time, and the severity factor.

Run	Temperature	Time	Severity Factor	Run	Temperature	Time	Severity Factor
#	°C	min	log R_0_	#	°C	min	log R_0_
1	0	0	0.00	11	160	15	2.94
2	120	10	1.59	12	160	20	3.07
3	120	15	1.76	13	170	10	3.06
4	130	10	1.88	14	170	15	3.24
5	130	15	2.06	15	180	10	3.36
6	140	10	2.18	16	180	15	3.53
7	140	15	2.35	17	190	10	3.65
8	150	10	2.47	18	190	15	3.83
9	150	15	2.65	19	200	10	3.94
10	160	10	2.77	20	200	15	4.12

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
