# Peer review of "Maize Silage Pretreatment via Steam Refining and Subsequent Enzymatic Hydrolysis for the Production of Fermentable Carbohydrates"

_molecules, 2020, doi:10.3390/molecules25246022_

Round 1

Reviewer 1 Report

The all year availability of material for bioenergy is a valuable issue. Ensiling the plant material is one option that the authors evaluate.

Materials:

Freezing is a possibility when material is used for research but will be less feasible if for bioenergy use. Would it be possible to comment on how the authors look at the storing of the raw ensiled material?

Line 130/131; the acid insoluble residue is the same as lignin. Where would the cellulose be?

It is not clear how the cellulose amounts are calculated.

Line 148; 50 um mesh is quite large and will allow particle to pass. What is the reason for choosing this size.

Table 1; It is not clear why 20 minutes treatment with 160 degree only?

Line 197; please specify the activity of the enzymes.

Table 3; please help the reader more. What is the be seen as lignin? The headline starch content – is this where the starch has been removed by enzymes? Then the title is not clear.

Table 6; what is CH. What is the % referring to?

Line 423; - war should be raw?

And a general question -why where the research not performed with non ensiled material as comparison?

Reviewer 2 Report

The manuscript titled “Maize Silage Pretreatment via Steam Refining and subsequent Enzymatic hydrolysis for the Production of Fermentable Carbohydrates” described enzymatic hydrolysis of the solid fiber following the steam refining experiment showing several beneficial results in the aspect of feedstock for lignocellulosic biorefineries. The raw materials have been treated from commercial maize chopper, followed by the ash content, starch content, carbohydrate detection. The pretreatment process was claimed to be important for good enzymatic hydrolysis. The manuscript is well presented and organized.

There are a few suggestions regarding the manuscript:

In section 3.2, the authors mentioned that the present findings may help reduce the operational costs, it is suggested that the author provide more information regarding how the presented findings can further reduce the operational costs.

The manuscript is suggested to add the information regarding how long the whole process takes and how it compared with other methods in the aspect of time-taken.
